# An Intensive Longitudinal Assessment Approach to Surveilling Trajectories of Burnout over the First Year of the COVID Pandemic

**DOI:** 10.3390/ijerph20042930

**Published:** 2023-02-08

**Authors:** Cynthia X. Pan, Robert Crupi, Phyllis August, Varuna Sundaram, Allison A. Norful, Joseph E. Schwartz, Andrew S. Miele, R. Rhiannon Simons, Emilia E. Mikrut, Elizabeth Brondolo

**Affiliations:** 1NewYork-Presbyterian Queens Hospital, Flushing, NY 11355, USA; 2Department of Medicine, Weill Cornell Medical College, New York, NY 10065, USA; 3Department of Surgery, Weill Cornell Medical College, New York, NY 10065, USA; 4School of Nursing, Columbia University, New York, NY 10032, USA; 5Renaissance School of Medicine, Stony Brook University, Stony Brook, NY 11794, USA; 6Department of Psychology, St. John’s University, New York, NY 11439, USA

**Keywords:** COVID-19 pandemic, emergency preparedness, burnout, caseload, frontline clinician, well-being, health care provider

## Abstract

Frontline clinicians responding to the COVID-19 pandemic are at increased risk of burnout, but less is known about the trajectory of clinician burnout as caseloads increase and decrease. Personal and professional resources, including self-efficacy and hospital support, can attenuate the risk of burnout. Yet, empirical data documenting how burnout and resources changed as the pandemic waxed and waned are limited. This intensive longitudinal prospective study employed ecological momentary assessment methods to examine trajectories of burnout and resources over the pandemic’s first year in a New York City hospital. A 10-item survey was emailed every 5 days to frontline clinicians (physicians, nurses, and physician assistants). The primary outcome was a single-item validated measure of burnout; predictors included daily hospital COVID-19-related caseloads and personal and professional resources. Clinicians (*n* = 398) completed the initial survey and an average of 12 surveys over the year. Initially, 45.3% of staff reported burnout; over the year, 58.7% reported burnout. Following the initial COVID peak, caseloads declined, and burnout levels declined. During the second wave of COVID, as caseloads increased and remained elevated and personal and professional resource levels decreased, burnout increased. This novel application of intensive longitudinal assessment enabled ongoing surveillance of burnout and permitted us to evaluate how fluctuations in caseload intensity and personal and professional resources related to burnout over time. The surveillance data support the need for intensified resource allocation during prolonged pandemics.

## 1. Introduction

Recent studies suggest that upward of 60% of the healthcare workforce reported some level of burnout or adverse psychological health during the COVID-19 pandemic [1,2]. Burnout occurs when individuals no longer perceive themselves as having sufficient resources to respond to work demands [3]. Burnout can manifest as emotional exhaustion, cynicism, detachment, and a loss of self-efficacy [3,4]. Burnout presents significant threats to both patients and providers, and has been associated with reduced patient safety [5,6], poorer treatment outcomes [7,8,9,10], increased patient mortality rates [11], reduced clinician productivity and satisfaction [12,13,14], and increased risk for clinician mental health symptoms [15,16].

The COVID-19 pandemic has exacerbated threats to clinical staff well-being and mental health [17] and increased the risk of burnout [18]. Frontline clinicians responding to the pandemic faced unprecedented threats, including high caseloads and risk of infection. Clinicians faced a deficit of resources, including an initial lack of evidence-based guidelines and effective therapeutics [18,19]. In response to these challenges, the Institute for Healthcare Improvement and the National Academies of Sciences, Engineering, and Medicine called for urgent attention to provider well-being and the prevention and treatment of burnout [20].

Using different measurement instruments and different criteria for defining burnout, most, although not all [21,22] survey studies indicate that 50% or more of clinical staff responding to the COVID-19 pandemic reported symptoms of burnout [23,24,25,26,27]. However, the existing data have largely been obtained using cross-sectional designs [22,23,24,25,26,27]. Cross-sectional evaluations in the early stages of a crisis can yield evidence of acute stress responses, but these studies provide limited evidence about the duration of these stress responses [28]. Alternatively, delayed evaluation—postponed until the crisis has passed—can also affect estimates of ongoing mental health risks. The post-crisis state can influence estimates of burnout during the crisis, due to retrospective recall bias [29]. Most critically, cross-sectional designs limit the ability to assess the course of burnout as the demands of the pandemic wax and wane.

During the first year of the pandemic, there were dramatic fluctuations in caseloads and other job demands. Caseloads rose rapidly during the first wave of the pandemic occurring in New York City from March through June 2020. During the summer, caseloads decreased, and then rose again during the second wave which started in the late fall. There were substantial variations in caseload from week to week through the first two waves. Yet, to our knowledge, there are no existing data examining if variations in caseload are associated with changes in burnout.

Intensive longitudinal assessments of the relations between caseload and burnout over time can help determine if burnout persisted across waves of the pandemic or if frontline clinicians recovered at least partly as caseloads declined between waves. These data can also help determine the causal direction of the relation between stress exposures (i.e., caseload fluctuations) and burnout and may clarify the timing of the relationship between stress exposure and burnout. The timing of exposure may matter as prior research indicates that some staff reports more mental health symptoms following versus during the acute crisis phase [25].

The evaluation of burnout or other mental health symptoms during a crisis is difficult as providers are engaged in critical care, and methods for ongoing assessment of the prevalence and course of burnout during the COVID-19 pandemic are lacking. Organizational psychologists have employed intensive longitudinal assessment methods to understand the relationship between work stress and well-being. A systematic review has documented that greater work intensity is associated with lower levels of well-being both within and across days [30]. However, to our knowledge, these methods have not been applied to study how fluctuations in work intensity over the course of the COVID-19 pandemic were associated with changes to healthcare professionals’ well-being. 

Data on the effects of acute changes in workplace demands (i.e., acute increases or reductions in caseload) on clinical staff burnout can guide clinicians, administrators, and policymakers as they make decisions about resource allocation. In the current study, ongoing surveillance of clinical staff well-being facilitated the integration of the findings into real-time planning efforts. These longitudinal methods are consistent with recommendations of the National Academies of Science [9] which highlight the importance of monitoring burnout over time to guide the design and timing of interventions to protect clinician well-being.

Prior research suggests the possibility of differences among nurses, trainees, and physicians in burnout, both in response to the pandemic [31] and during their regular clinical responsibilities [32,33]. Some data suggest that nurses reported more burnout during the pandemic than doctors [31], possibly because they had greater day-to-day responsibilities for providing direct care for suffering patients. However, to our knowledge, no studies have examined professional role differences in the course of burnout during the pandemic. Intensive longitudinal assessments can allow us to determine if nurses had higher levels of burnout than physicians or other professional groups during times when caseloads rose and if they also recovered more slowly when caseloads fell.

Burnout may be prevented and/or mitigated by critical psychosocial resources including social support [34], perceived self-efficacy [35], the meaning of work [36], and opportunities for professional development [37]. Yet, high levels of work stress can undermine these resources. To date, it has not been clear how these resources were affected by the different levels of demand presented during the pandemic. An intensive longitudinal design may permit us to examine the degree to which resources were replenished following the first wave of the pandemic but were depleted again during the second wave. Further analyses can examine if variations in psychosocial resources predicted changes in burnout.

The present analyses address three questions: (1) How much did burnout change over the course of the year, and how sensitive was burnout to changing caseloads and other hospital conditions? (2) Were the trajectories of burnout the same across professional roles (e.g., across nurses, attending physicians, trainees, and advanced practice clinicians)? (3) Was the presence of personal and professional resources associated with lower levels of burnout, and did these resources change over the year?

## 2. Methods

### 2.1. Transparency and Openness

To foster transparency, we describe the sampling plan, all measures and all data exclusions. All analyses were performed with SAS 9.4 (SAS Institute). All measures and programming are available at OSF: (https://osf.io/fe2rz/?view_only=477156be00c9470781d629cc6c6ce083, accessed on 14 July 2022). Data are available by contacting the corresponding author.

The study was not pre-registered as it was developed as a rapid response to the pandemic and was conducted to facilitate hospital planning. The proposal was approved by the Institutional Review Boards of both university and hospital settings and all participants provided consent. Reporting follows STROBE criteria [38].

### 2.2. Overview

We implemented an intensive longitudinal assessment in which we collected data on caseload, burnout, and resources every five days as the pandemic unfolded, a strategy which permitted us to understand relations among variables without incurring some of the difficulties associated with retrospective surveys [29,39]. We examined trajectories of burnout over the first year of the COVID-19 pandemic and investigated how changes in hospital caseloads over the first two waves of the pandemic were related to changes in burnout.

The current longitudinal study employed a brief survey to assess clinical staff well-being and burnout in real-time throughout the first year of the pandemic (April 2020–March 2021). This study was conducted in a community teaching hospital in New York City (NYC). In the early stages of the pandemic, from March to May 2020, NYC was the epicenter of the COVID-19 pandemic. NYC experienced a second wave in the late fall of 2020, which extended into the winter of 2021. We used an intensive longitudinal prospective design using EMA methods. We administered surveys every 5 days via email from 14 April 2020 to 31 March 2021, resulting in 70 administration points over the year. We chose to administer surveys every five days to permit us to capture the effects of rapid changes in caseloads while avoiding day-of-the-week effects, which would have occurred if we assessed participants every 7 days. Data were collected at a single site, an academic-affiliated community medical center in Queens, NY, USA. Data collection was initiated at the height of the pandemic in New York City. This survey was disseminated hospital-wide to all frontline clinicians in the hospital workforce. There were no exclusion criteria. The survey was part of a hospital-wide effort to surveil workforce burnout.

The project emerged from a University-Hospital partnership. Faculty and students from the University created the survey with input from hospital staff, administered the survey through Qualtrics electronic survey software and collected and analyzed all data. The University’s role in data collection and management protected the confidentiality of the hospital employees. Physicians, nurses, and administrators from the hospital provided access to hospital personnel and guided the development of the survey, interpretation of the data, and timely dissemination of results to hospital leadership and employees. Six undergraduate and four graduate students were responsible for uploading and distributing the Qualtrics survey, managing the data, tracking caseloads, developing tables for the manuscript, and conducting literature reviews.

### 2.3. Participants and Recruitment

Eligible participants included all 2023 hospital-based clinical staff (i.e., physicians, nurses, and physician assistants). Participation was voluntary and no incentives were offered. Department heads sent an email detailing the study’s purpose, a link to the consent and the survey. The survey was named “Frontline Clinician Well-Being Study”. Participants received daily reminders to complete the initial survey. Follow-up surveys were sent every five days for up to 1 year following the completion of the initial survey. Participants received two reminders via email to complete each follow-up survey.

### 2.4. Survey Measures

The full Frontline Clinician Well-Being survey is included in Appendix B. Seven questions assessing demographic and professional information (i.e., sex, age, race, marital status, professional role, years of experience, and department) were administered in the initial survey. Ten questions addressing clinical responsibilities, personal and professional resources, and burnout were administered at every assessment point and are described below. Items were necessarily limited and brief to reduce participant burden over the course of repeated administration. The personal resources/self-efficacy measure has been employed in a study of psychological resources among Italian healthcare providers [40]. Available single-item measures of professional development and meaning of work were not available and were developed based on the literature.

***Clinical Responsibilities.*** Two items assessed clinical responsibilities, inquiring if participants had worked in a clinical capacity within the past 24 h and if they were currently working outside their usual professional role.

***Personal Resources.*** Four items assessed self-efficacy along four dimensions: energy, cognitive capability, emotion regulation capacity, and self-care capacity. These items were developed for this project but were modeled on concepts drawn from other measures [41,42]. The items had a Cronbach’s alpha of 0.79 in this sample.

***Professional Resources.*** Two items assessed crisis-related professional growth and perceptions of meaningful work. These items were treated separately as the relations among the items fell short of conventional acceptability for Cronbach’s alpha.

***Hospital Support.*** One item inquired about perceptions of the provision of timely information and support from the hospital.

***Burnout.*** Burnout was assessed with the Dolan et al. [43] single-item validated measure of burnout in healthcare workers (see Appendix B). Responses on a 5-point ordinal scale ranged from “I enjoy my work. I have no symptoms of burnout” to “I feel completely burned out and often wonder if I can go on. I am at the point where I may need some changes or may need to seek some sort of help”. According to the Dolan et al. [43] measure, a score of one indicates no burnout, a score of two indicates the experience of stress but not burnout, a score of three indicates an acknowledgment of burnout and one or more symptoms of burnout, a score of four indicates persistent burnout, and a score of five indicates severe burnout. Participants were regarded as reporting burnout if their scores were three or above, according to established criteria [44,45].

We selected this single item to reduce the length of the survey and reduce participant burden. Researchers have demonstrated that when this single item is compared to the emotional exhaustion scale item on the Maslach Burnout Inventory, the measures are highly correlated (0.79) and the Dolan measure shows a sensitivity of 83.2% and specificity of 87.4% [43].

***Needed Resources.*** Every fourth survey, participants were asked to review a list of resources they might need to respond to the pandemic and maintain well-being. The list was generated by the research team based on interactions with clinical staff. Participants were asked to indicate if obtaining this resource was a low priority or a high priority. Results of the analyses of these items are included in Appendix A.

***Caseload.*** Five-day moving averages of COVID caseload were calculated using Proc Expand from SAS 9.4. Over the year, data on caseloads were not available for a total of 43 (episodes) periods lasting an average of 4.63 days (range 1 to 22 days), generally during periods in which caseloads had declined substantially. For the brief periods in which caseload data were unavailable, we interpolated caseload values calculating replacement values based on the last date data were available and the next date data were available.

### 2.5. Data Analysis

Descriptive statistics were used to evaluate participation rates. Chi-square analyses examined differences between those who participated in the study only once versus those who participated more than once. Predictors of burnout category (i.e., no reported burnout, any reported burnout) during the initial survey administration were evaluated with logistic regression (Proc Logistic, SAS 9.4). Predictors included sex and professional role. Professional role was recoded into four categories (physicians, Graduate Medical Education trainees (i.e., residents and fellows), nurses and nurse practitioners (NPs), and physician assistants (PAs). The date of initial survey completion was included as a categorical predictor to control for differences in overall conditions, including caseload. To evaluate the degree to which participants experienced sustained burnout over time, we calculated the proportion of assessment points following the initial assessment in which the participant reported a burnout score of 3 or more (i.e., minimum score for identifying burnout [43]).

To examine longitudinal changes over time in caseload, burnout, and the personal and professional resources associated with burnout we used multilevel mixed logistic regression analyses (MMLM) (Proc Glimmix, SAS 9.4). MMLM was employed in these analyses as these models are more robust to missing values than traditional regression analyses [46].

Time was treated as a continuous variable with a one-point change corresponding to a period of 5 days. In this coding, 0 = 14 April 2020, the date the study started, 0.2 = 15 April 2020, 1 = 20 April 2020, and so on. Instead of assuming that burnout (or resources) exhibits a linear trajectory over time, we estimated a B-spline model. The B-spline models the relationship of the outcomes to time as a smooth curve (cubic function), allowing the parameters to shift at specific times (called “knots”) subject to the constraint that the curve maintains its smoothness pre- to post-knot. The complexity (versus overall smoothness) of the curve is determined by the number of knots. The Bayesian Information Criterion (BIC) statistic was used to determine the minimum number of knots required to arrive at a parsimonious, good-fitting curve. Inclusion of the interaction of professional role × B-spline (time) fit a separate time trajectory for each professional role and provided a test of whether the pattern of change over time differed by professional role.

To understand trends over time, we compared average burnout (or resource) levels for 3 periods: (1) the initial wave of the pandemic (i.e., April through June 2020); (2) the Summer-Fall when caseloads dropped (i.e., August–October); and (3) the peak of the second wave when caseloads increased again (i.e., January through March 2021). From a model in which the primary predictor was month, rather than continuous time or time (B-spline), contrasts were used to estimate and test the difference between two periods, comparing the average burnout level for the months of one period to those of another period.

## 3. Results

### 3.1. Response Rate and Personal and Professional Correlates of Response

A total of 398 clinicians participated in the initial Frontline Clinician Well-being survey administration yielding a response rate of 19.7% (Table 1). Five participants completed only the demographics portion of the initial survey, and only 373 of the 398 participants completed the burnout question on the initial survey. The survey was administered every five days for a total of 70 administrations over the course of the year. Most participants (*n* = 322, 80.9%) completed the survey more than once. An average of 72.6 participants completed the survey at each administration point (range = 26–383 participants, median = 52). On average, participants completed 12.7 surveys (range = 1–70, median = 6), Repeat administrations permitted the collection of 5070 surveys over the year. The median time to completion of the survey was 54 s; 75% of completed surveys were finished within 104 s.

Participation more than once (versus only once) was unrelated to demographic characteristics (i.e., age, gender, or race), professional role or department or initial burnout level (all *p* values = NS). The number of surveys completed per participant, an index of participation over time, was unrelated to the average burnout score (*r*(373) = −0.05, *p* = NS). All professional roles were represented in the study. In the initial survey, there were roughly equal response rates across professional roles (Table 1). However, there were professional role differences in participation over time (*F*(3397) = 4.87, *p* = 0.0025). Attending physicians completed more surveys over time than nurses or trainees (corrected *p* < 0.05) but did not differ from physician assistants.

Sociodemographic differences in burnout are presented in Table 2. Logistic regression analyses indicated that on the initial survey, women were more likely to report burnout than were men; younger participants were more likely to report burnout than were older participants; trainees and nurses were more likely to report burnout than were attending physicians, and those working outside of their usual role were more likely to report burnout than those working in their usual role.

### 3.2. Initial and Subsequent Burnout Rates

On the initial survey, just over half (*n* = 204, 54.7%) did not report symptoms of burnout (i.e., scores 1 and 2); 121 (32.4%) reported moderate burnout (score 3); and 48 [12.9%) reported high levels of burnout (scores 4–5). Over the course of the year, 58.7% (*n* = 277) reported burnout at least once. Many of those who reported experiencing burnout in the initial survey and who completed additional surveys (*n* = 136) repeatedly experienced burnout. These participants reported burnout on an average of 64% (*SD* = 32%, range = 2–100%) of their remaining assessments; 61.03% of these participants reported experiencing burnout on half or more of their remaining responses.

### 3.3. Caseload and Burnout

Caseloads varied over the year (Figure 1). For two weeks prior to survey initiation, the COVID-19 census peaked at 597 and exceeded the hospital capacity of 535 beds. At the study onset (14 April 2020), there were 480 patients with COVID-19. From August through October 2020, caseloads diminished and remained low, averaging less than 40 COVID patients per day. A second wave began in December 2020, with COVID-19 caseloads ranging from 171 to 230 from January through March 2021. Caseloads during the second wave were not as high compared to the start of the pandemic, but they sustained around 200 patients from January to March. The overall hospital census, however, returned to at or above the maximum hospital capacity.

There was a significant association of caseload (B-spline) to burnout (*F*(3, 1746) = 63.26, *p* < 0.001) over the course of the year. There was also a significant association of time (B-spline) to burnout (*F*(5, 1028) = 46.57, *p* < 0.001). Burnout initially decreased during the summer as caseloads decreased, and then increased again in the late Fall/Winter as caseloads increased (Figure 2). Analyses of changes in effect sizes indicated that there was a small, but significant decrease in burnout levels from Wave 1 (i.e., April–June 2020: *t* = −5.29, *p* < 0.001, *d’*=−0.19) to the period during Summer/early Fall (i.e., August through October 2020), when caseloads also decreased. There was a significant increase in burnout from Wave 1 to Wave 2 (January–March 2021: *t* = 6.77, *p* < 0.001, *d’*= 0.30).

### 3.4. Professional Role Differences in Burnout

In the initial survey, nurses were more likely to report burnout than attending physicians, even after controlling for sex (Table 2). There were significant differences by professional role in the course of burnout over time (B-spline) (*F*(15, 1071) = 3.18, *p* < 0.001) (Figure 3), and effects remained significant controlling for sex (*p* < 0.001). Nurses had higher levels of burnout during Wave 1 (Estimate = 0.387, *SE* = 0.10, *t* = 3.61, *p* < 0.001, 95% *CI* = 0.172–0.582), and despite recovering more quickly than attending physicians during the first wave, they had higher levels of burnout during Wave 2 (Estimate = 0.506, *SE* = 0.126, *t* = 4.03, *p* < 0.001, 95% *CI* = 0.259–0.752). In contrast, trainees had higher levels of burnout than attending physicians at Wave 1 (Estimate = 0.447, *SE* = 0.140, *t* = 3.21, *p* < 0.01, 95% *CI* = 0.173–0.723), but not at Wave 2 (Estimate = 0.291, *SE* = 0.224, *t* = 1.30, *p* = NS, 95% *CI* = −1.48–0.730).

### 3.5. Participants’ Personal and Professional Resources and Burnout

We conducted four separate mixed model logistic regression analyses to examine the associations of self-efficacy, opportunities for professional development, meaningful work, and perceptions of hospital support to burnout over the course of the year. Higher levels of the B-spline of each resource were associated with lower levels of burnout (all *p* values < 0.0001). When all four types of resources were entered simultaneously, self-efficacy (B-spline): (*F*(4, 4392) = 134.28, *p* < 0.001), professional development (B-spline): (*F*(4, 4417) = 8.72, *p* < 0.001), and hospital support (B-spline): (*F*(4, 4334) = 27.68, *p* < 0.001) were each independently significantly associated with burnout over the year.

To examine changes to respondents’ perception of personal and professional resources over time, we compared changes in the responses to questions about these resources from the initial wave of the pandemic (April–June 2020) to the period corresponding to the second wave (January–March 2021) using the same analytic approach as we did for changes in burnout. From the first to the second wave of the pandemic, there were significant decreases in perceptions of self-efficacy (*t* = −6.52, *p* < 0.001: *d’* = −0.30); capacity for professional development (*t* = −9.81, *p* < 0.001, *d’* = −0.42); perceptions of meaningful work (*t* = −10.83, *p* < 0.001; *d’* = −0.50); and perceptions of hospital support (*t* = −10.58, *p* < 0.001, *d’* = −0.46).

## 4. Discussion

To obtain “real-time” assessments of the well-being of frontline healthcare workers at one of the busiest urban hospitals in Queens, New York, we conducted an intensive longitudinal assessment, administering an electronic survey every five days throughout the first year of the COVID-19 pandemic. Almost 20% of clinical staff hospital-wide completed the initial survey, despite high levels of work responsibilities. Participation rates were consistent with past voluntary burnout surveys in clinicians during non-crisis periods [47,48]. Continued participation over 12 months did not appear to be biased by initial levels of burnout or personal characteristics.

The longitudinal, prospective design permitted us to understand the costs of the sustained nature of the pandemic. The second wave was associated with significant declines in personal and professional resources, including self-efficacy and a sense of professional development. The experience gained during the earlier stages of the pandemic did not seem to buffer staff from the costs to their well-being when COVID cases rose again.

Even without mandatory participation or the use of incentives, the findings suggest that brief, electronically administered surveys can provide real-time information on clinician well-being useful for planning and guiding emergency operations. Real-time data collection and analysis enabled rapid dissemination of participants and hospital leadership. Throughout the first year, we conducted a series of meetings with different groups of hospital leaders (i.e., from the Department of Medicine, Nursing, and other groups) to disseminate the results. All participants and hospital leadership received three newsletters over the course of the first year of the pandemic, containing the findings from the study. Emergency planning efforts may consider using similar approaches, as they are consistent with the Surgeon General’s recent advisory report on the importance of addressing burnout in healthcare workers [49].

Hospital leaders may benefit from intensive longitudinal data. These data can facilitate drawing causal connections about the relations between work demands and clinician well-being. Empirical data can strengthen the argument for resource allocation from governmental and other sources. One of the most important goals of the study was to develop and test a data collection process that could provide sensitive and timely feedback about the effects of any intervention. The use of frequent quantitative assessments can aid Plan-Do-Study-Act (PDSA) approaches to improve clinical staff well-being and other quality improvement efforts [50]. University-Hospital partnerships can facilitate the process.

Consistent with other reports on the mental health effects of the pandemic on frontline clinical staff (e.g., [51]), the results of our survey indicated that burnout was prevalent and persistent. Almost 60% of the sample reported burnout during at least one assessment over the year. Those who reported burnout initially, during the peak of the crisis, were more likely to report burnout throughout the year, highlighting the need for early interventions.

Over the year, burnout levels decreased gradually as caseloads decreased. Burnout increased and exceeded Wave 1 levels as caseloads increased during Wave 2. Caseloads in the second wave were not as high as those during the first wave, but they remained elevated for several months. The sustained demand may have generated greater exhaustion and concerns about the future. However, additional demands from non-COVID-19 related illnesses or the re-introduction of postponed treatments may have offset the benefits of decreased COVID-19 caseloads during the early months of 2021.

The survey provided insight into those most at risk. Burnout was initially more common among nurses and trainee physicians than among attending physicians. Nurses reported higher levels of burnout than physicians throughout the year, but trainee physicians reported higher levels of burnout only in the initial wave of the pandemic. The literature has yielded mixed findings on the relative burnout scores of physicians versus nurses [2,33,52]. These findings suggest that longitudinal studies are valuable for answering questions about professional role differences in burnout, as there may be differences among professional groups both in acute stress responses as well as stress recovery over time.

Professional role differences in this sample may be a function of the type of emotional demands presented by managing COVID-19 patients. Nurses have a more direct role in the day-to-day management of patient needs, resulting in greater exposure to patients’ pain and suffering, potentially increasing the risk of burnout. Therefore, discipline-specific interventions to mitigate burnout may better support the nursing profession when they target roles, responsibilities and policies [2,53,54].

The study provided some insight into potential targets for intervention. Both personal resources, including self-efficacy, and professional resources, including perceptions of support from the hospital and a sense of professional development, were inversely associated with burnout. These findings are consistent with other reports which suggest support and a sense of mastery increase resilience to burnout [55,56]. Hospital administration may need to provide additional recognition and material support to clinical staff when a pandemic persists.

### Limitations

Study limitations include the use of a single site and convenience sampling, which likely affected estimates of the prevalence of burnout. The initial response rate was almost 20%, a rate consistent with prior studies of voluntary samples of clinicians in the workplace [47,48]. Most participants responded more than once, but the response rate declined over time. Importantly, repeated participation did not appear to be biased by the initial response to the survey. Further, although variations in the response rate over time resulted in missing data, the analytic method used provides valid estimates under the substantially less restrictive assumptions of missing at random (MAR) versus those associated with missing completely at random (MCAR) [57].

Further research will be needed to understand variations in participation over time. As the use of intensive longitudinal designs in the workplace is a relatively new approach, there is limited information on the proportion of participant retention needed at each time point to estimate changes over time in a reliable and unbiased manner (see [58] for a review of issues in longitudinal modeling). Participants in this study were all volunteers and no incentives were employed. It would be important to understand if participation rates over time would improve with less frequent testing (e.g., every 10 days) or with the use of incentives or other approaches. Additional research is needed to assess the EMA approach in other hospitals.

We did not measure vaccine uptake among clinical staff or patients. However, in NYC, the vaccine was not widely available for healthcare professionals until mid-December 2020 and was not widely available to the general public until March 2021 [59].

Data from the study were shared with leadership and participants beginning in July 2020, after the first Wave. We acknowledge that data sharing with administrative leadership and clinical staff may have potentially affected leader behavior and administrative decision-making, and consequently influenced the outcome of the study. It is important to note that this was pragmatic research, conducted with the aim of investigating the trajectory of clinician well-being during a significant health crisis. These data were the only real-time surveillance of burnout in the early stages of the outbreak. Sharing information with hospital leadership was a necessary component of the project to enable leadership to gain feedback on stress and burnout in employees. As one of the initial epicenters of the US COVID-19 pandemic, the workforce in this hospital experienced tremendous strain early in the pandemic, and it would have been unethical to measure the large scale of burnout and not share the information which could help to mitigate it. Understanding the effects of information sharing on burnout and well-being would be an important area for further research.

## 5. Conclusions

Our data suggest that the administration of a brief electronic survey during a pandemic may help hospitals conduct real-time surveillance of burnout among their healthcare workforce. Our study presents longitudinal evidence suggesting that burnout tracks workplace demands during a pandemic. However, personal and professional resources do not fully recover over time, as new demands replace those associated with the pandemic.

Health systems should consider workforce policies and interventions to counteract the adverse impact of burnout in real-time and over a prolonged period. The methods presented in this study may guide administrators, policymakers, and researchers with a strategy for real-time prospective measurement of burnout, among other adverse outcomes. The methods could be employed to conduct real-time and brief assessments of interventions to improve employee well-being.

## Figures and Tables

**Figure 1 ijerph-20-02930-f001:**
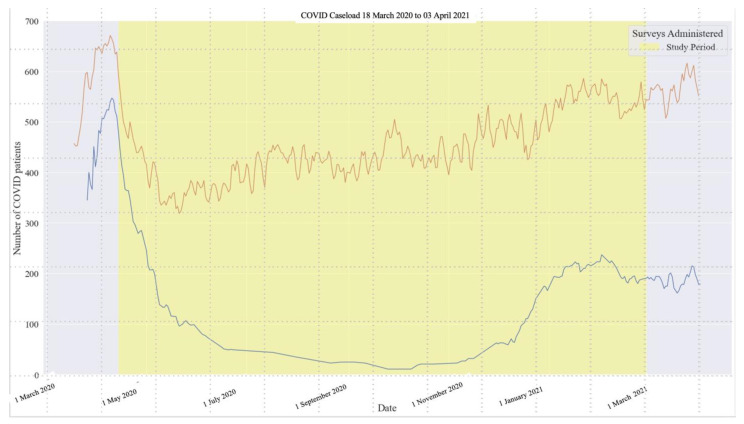
COVID Caseload and Total Hospital Census During Study Period. Note: The upper (brown) line depicts the total hospital census. The lower (blue line) depicts the COVID-19 caseload. The first wave occurred in NYC from March to June 2020, and the second wave began in December 2020.

**Figure 2 ijerph-20-02930-f002:**
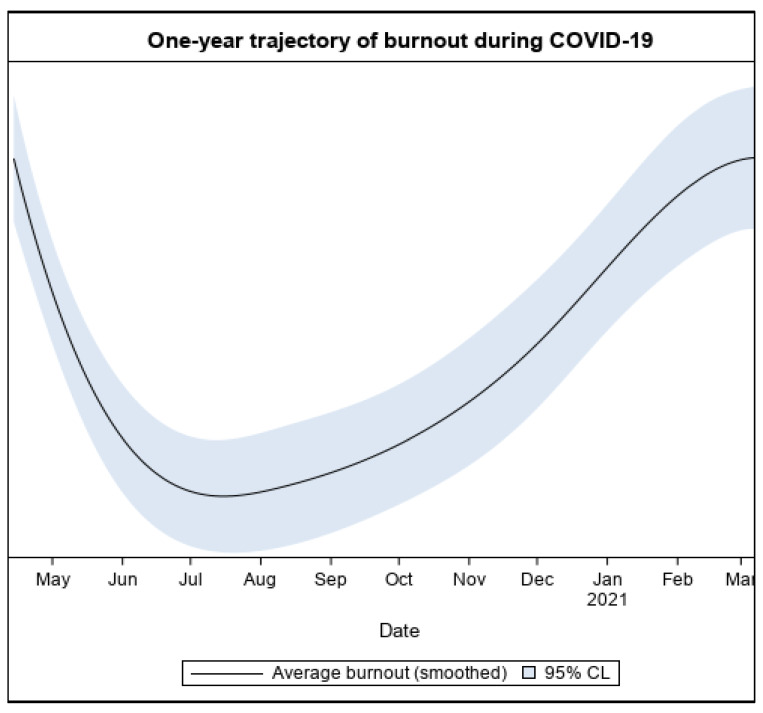
The one-year trajectory of burnout.

**Figure 3 ijerph-20-02930-f003:**
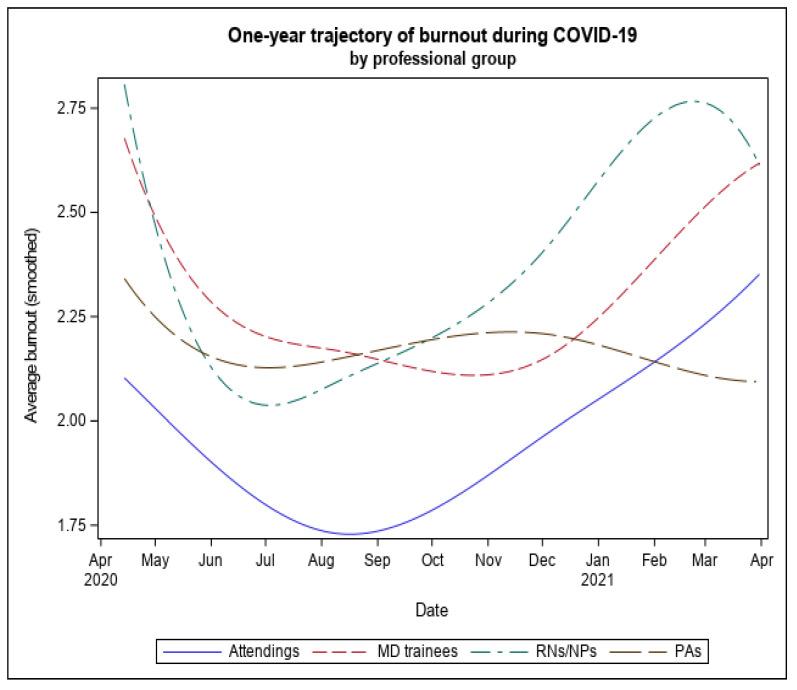
The one-year trajectory of burnout by Professional Role.

**Table 1 ijerph-20-02930-t001:** Initial Assessment: Participant characteristics for full sample and by professional role.

Variables	Full Sample	Attending Physicians	Residents and Fellows	Nurses and Nurse Practitioners	Physician Assistants
*n*, % of total who received the survey invitation	398 (19.7%) of the 2023 who received the survey invitation	86 (23.1%) of 373 Attending Physicians who received the survey invitation	61 (22.8%) of 267 Residents and Fellows who received the survey invitation (Residents: *n* = 57/251; Fellows: *n* = 4/16)	209 [17.8%) of 1177 Nurses and Nurse Practitioners who received the survey invitation (Nurse: *n* = 199/1144; Nurse Practitioner: *n* = 10/33)	42 (20.4%) of 206 Physician Assistants who received the survey invitation
Sex (*n* = 398) ^1^					
Men	121 (30.4)	49 (57.0)	32 (52.5)	26 [12.4)	14 (33.3)
Women	277 (69.6)	37 (43.0)	29 (47.5)	183 (87.6)	28 (66.7)
Age (*n* = 388) ^1^					
<34	149 (38.4)	15 [17.9)	52 (86.7)	62 (30.5)	20 (48.8)
35–49	157 (40.5)	42 (50.0)	8 [13.3)	91 (44.8)	16 (39.0)
50–59	47 [12.1)	15 [17.9)	0 (0)	30 [14.8)	2 (4.9)
60+	35 (9.0)	12 [14.3)	0 (0)	20 (9.9)	3 (7.3)
Race and Ethnicity (*n* = 387) ^1^					
White	145 (37.5)	43 (51.2)	22 (36.7)	62 (30.7)	18 (43.9)
Chinese	58 [15.0)	15 [17.9)	10 [16.7)	24 [11.9)	9 (22.0)
Black	40 [10.3)	2 (2.4)	3 (5.0)	34 [16.8)	1 (2.4)
Asian Indian	36 (9.3)	13 [15.5)	8 [13.3)	13 (6.4)	2 (4.9)
Latinx	30 (7.8)	3 (3.6)	1 [1.7)	19 (9.4)	7 [17.1)
Other ^2^	78 (20.2)	8 (9.5)	16 (26.7)	50 (24.8)	4 (9.8)

Note. Of the 398 participants, 373 completed the question on burnout. ^1^ *n* = the number of participants who provided information on this variable. ^2^ The “Other” group included those who identified as Filipino, Vietnamese, Korean, Japanese, Other Asian, Other Pacific Islander and some other races. Each of these groups represented less than 8% of the sample.

**Table 2 ijerph-20-02930-t002:** Initial Assessment: Odds ratios (OR) and proportions of participants reporting burnout by demographic and professional role categories.

Group	N (%) Reporting Any Burnout	OR, 95% CI ^1^
Sex ^a^ (*n* = 373)		
Men	33 (29.2)	0.38 (0.24–0.61)
Women	136 (52.3)	
Age ^b^ (*n* = 358)		
<35 years	77 (54.6)	4.40 [1.78–10.87)
35–49 years	63 (44.4)	2.83 [1.15–6.97)
50–59 years	13 (30.2)	1.61 (0.55–4.66)
60 or more years	7 (21.9)	
Professional Role ^c,d^ (*n* = 373)		
Trainees (Residents and Fellows)	26 (44.83)	2.16 [1.06–4.42)
Nurses and Nurse Practitioners	109 (55.61)	3.32 [1.89–5.86)
Physician Assistants	12 (31.58)	1.22 (0.53–2.84)
Attending Physicians	22 (27.16)	-
Department of Medicine ^e^ (*n* = 357)		
Yes	55 (50.0)	1.01 (0.99–1.03)
No	106 (42.91)	-
Out of Role ^f^ (*n* = 371)		
Yes	78 (53.06)	1.75 [1.14–2.68)
No	90 (40.18)	-

Note. ^1^ Odds ratios obtained by logistic regression. ^a^ Percentages reflect the proportion of individuals within each sex category who report burnout. ^b^ Percentages reflect the proportion of individuals within each age group who report burnout. ^c^ All significant comparisons remained significant controlling for sex. ^d^ Percentages reflect the proportion of individuals within the professional role who report burnout. ^e^ Percentages reflect the proportion of individuals within the Department of Medicine who report burnout. ^f^ Percentages reflect the proportion of individuals redeployed who report burnout.

## Data Availability

The data presented in this study are available upon request from the corresponding author.

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
