# Peer review of "An Intensive Longitudinal Assessment Approach to Surveilling Trajectories of Burnout over the First Year of the COVID Pandemic"

_ijerph, 2023, doi:10.3390/ijerph20042930_

Round 1

Reviewer 1 Report

Dear authors,

Thank you for giving me the opportunity to review your manuscript. This is a well-designed study that is worthy of publication barring a few small ammendments.

Introduction

Well written and introduces concepts in burnout well and summarises the issues pertaining COVID endemicity and its effects on the mental wellbeing of healthcare workers.

Consider including DOI: 10.1016/j.jamda.2022.01.059 as it is also a foray into the effects of COVID endemicity on burnout.

Methods

Lines 187-194: What are the Cronbach alphas for the Burnout parameter? What were the reasons why this was selected as opposed to other burnout inventories such as Maslach's and Oldenbrurg? I understand it could be because brevity was a key priority and also because that paper by Dolan et al demonstrates a good correlation a single-item version of Maslach as opposed to the multiple item version.

Results

Lines 246: What does "an average of 72.6 participants completing the survey at each administration..." mean? 

Lines 296-301: What about non-COVID caseload. While there is a drop in COVID cases, there was a well-documented and well known rise in non-COVID cases, largely to clear the backlog of elective cases. Is this data i.e. the non-COVID cases available?

Figure 1 & 2: Is is possible to overlap the COVID cases with the trajectory in burnout to better demonstrate the trend

Figure 2: It woudl be good to mention that burnotu refers to >= 3 points

Discussion

Lines 370-80: This is a strong point i.e. that a similar brief studies would be useful for keeping track on burnout over a long period of time

Lines 396-404: How do changes in non-COVID cases relate to this?

Appendix A: Are components of Page 15 i.e. "Think about what you neded to improve you well-being and performance. Check all that apply:" reported in your manuscript.

I feel that if the above issues are addressed, this manuscript would be suitable for publication. I looked forward to receiving a revision of your manuscript. 

Author Response

Dear reviewer,

Thank you for your efforts and time for review. We have revised manuscript according to your comments.

Kind regards

Reviewer 2 Report

Title: An Intensive Longitudinal Assessment Approach to Surveilling Trajectories of Burnout Over the First Year of the COVID Pandemic (IJERPH-2139351)

The authors by this intensive longitudinal prospective study employed ecological momentary assessment methods to examine trajectories of burnout and resources over the pandemic’s first year in a New York City hospital. 

The manuscript is well written and well supported by consistent references. The conclusions are largely sound and improve the existing knowledge. 

Few suggestions to improve the manuscript are reported below.

Lines 80-87 e 119-129 The authors should transfer these sentences to the material and methods paragraph. They should conclude the introduction by describing the objectives,  of their research.

Author Response

Dear reviewer,

Thank you for your efforts and time in reviewing. We have revised the manuscript according to your comments.

Kind regards,

All authors

Reviewer 3 Report

Dear authors, 
comments on you article are below.

In the introduction the use of the Intensive Longitudinal Assessment Approach was justified several times as being able to follow the burnout trend in the considered sample over time. It would also be interesting to point out the differences between the other burnout measurement tools and the advantages related to the use of the chosen instrument. 

There were not any refer to the adopted COVID-19 pandemic management policies althought they influenced deeply the working conditions of the sample. It moreover should justify the selection of the considered sample, than does not emerge both in the introduction and in the method. I think the initiation of vaccination had a strong impact in the psychological sphere of the individuals. 

Then, reference was made to student involvement in the  structuring of the survey but then  it is not explained who they are, how many they are and how they were involved.

In method, the description of survey reported  it is understood how some questions used have already been validated and others have not. Perhaps it would be useful to better justify this choice. Here the meeting with hospital leaders must be presented although the description of this actions should be weighed (seen the comments in about dicussion) 

In results the B-spline could be introduced after the presentation also of the direct obtained data. Then, the some results could be presented also in association with the structure of the survey. 

In discussion, significant adjustments are required. In my opinion the sharing of the partial results with leaders introduced a significant bias in the research. In fact leaders were able to act in the working enviromental of the sample modifying thier experience and, consequently, their levels of burnout. So all the following considerations were misleading.
I could agree with authors about the important informative contribution of the submitted survey for leaders but there is not a strong relationship with resorces allocation or policies making. The discussion should focus on the improvement of competence and abilities of health professionals. 

Finally, the conclusions should be modified according to the suggestions given. 

Author Response

(The authors gave the same response as above.)

Reviewer 4 Report

ABSTRACT

The research topic is of scientific and social interest.

There is cohesion in the article, between the section on Theoretical Framework and the subsequent sections, which describe the study and draw conclusions, have a correct, well-structured and cohesive design.

In general, the article is correct and I consider that the topic is in line with the journal’s research objectives.

INTRODUCTION:

The study objective is well defined and identified in both the abstract and the introduction.

The subject under investigation is of growing scientific and social interest. The investigation is current.

The result is a work that can be the basis for many others in this field.

The statistical technique used is well justified and explained, both the process and the results, despite not being widely used in these investigations, which implies an added effort by the authors of the article.

MATERIALS, METHODS and RESULTS:

The statistical treatment is correct.

DISCUSSION and CONCLUSION:

The conclusions are well drawn and interesting.

The discussion is correct.

Author Response

(The authors gave the same response as above.)

Round 2

Reviewer 3 Report

Dear authors, 

I noted that the suggested corrections have all been accepted. 
Given the significance of the article topic, these were made with the intention of consolidating and refining the article so that it can hopefully be an important contribution to the analysis of the impact of COVID-19 pandemic on healthcare organisations at worldwide level. 

Now, the article describes also methodological aspects, that are useful for possible replication in other contexts or in other periods. Another possibile developments of the research could be the integration of the wide literature on the topic, also using so some systematic reviews. 

However, the bias given by the involvement of hospital managers remains. For the exploratory nature of the study and the more detailed description of the method, it is believed that the study can be published in this form.

Best regards